# Photosynthesis, Respiration, and Growth of Five Benthic Diatom Strains as a Function of Intermixing Processes of Coastal Peatlands with the Baltic Sea

**DOI:** 10.3390/microorganisms10040749

**Published:** 2022-03-30

**Authors:** Lara R. Prelle, Ulf Karsten

**Affiliations:** Applied Ecology and Phycology, Institute of Biological Sciences, Albert-Einstein-Strasse 3, University of Rostock, D-18057 Rostock, Germany; ulf.karsten@uni-rostock.de

**Keywords:** growth rate, flooding, ecocline, climate change, temperature, heterotrophy

## Abstract

In light of climate change, renaturation of peatlands has become increasingly important, due to their function as carbon sinks. Renaturation processes in the Baltic Sea include removal of coastal protection measures thereby facilitating exchange processes between peatland and Baltic Sea water masses with inhabiting aquatic organisms, which suddenly face new environmental conditions. In this study, two Baltic Sea and three peatland benthic diatom strains were investigated for their ecophysiological response patterns as a function of numerous growth media, light, and temperature conditions. Results clearly showed growth stimulation for all five diatom strains when cultivated in peatland water-based media, with growth dependency on salinity for the Baltic Sea diatom isolates. Nutrient availability in the peatland water resulted in higher growth rates, and growth was further stimulated by the carbon-rich peatland water probably facilitating heterotrophic growth in *Melosira nummuloides* and two *Planothidium* sp. isolates. Photosynthesis parameters for all five diatom strains indicated low light requirements with light saturated photosynthesis at <70 µmol photons m^−2^ s^−1^ in combination with only minor photoinhibition as well as eurythermal traits with slightly higher temperature width for the peatland strains. Growth media composition did not affect photosynthetic rates.

## 1. Introduction

The Baltic Sea in northern Europe is almost entirely enclosed by land with a catchment area of 1.74 million km^2^ [1], and a coastline that is typically shaped by grassland, marches, peatlands, and inhabited areas. Depending on the meteorological conditions, only small exchange processes exist with the adjacent North Sea. In addition, considering the freshwater riverine discharge, this leads to a horizontal salinity gradient in the Baltic Sea ranging between <3 S_A_ in the north and approximately 27 S_A_ in the south-east [2]. Being located in the temperate zone, abiotic factors such as temperature and light availability in and around the Baltic Sea exhibit strong seasonality.

In recent decades, the influence of climate change has become more concrete leading to new environmental conditions for the Baltic Sea. As a consequence of glacial melting and thermal expansion [3], the sea level of the Baltic Sea is continuously increasing by 2 mm yr^−1^ in the southern Baltic Sea [4], which is enhanced by the glacio–isostatic subsidence of the southern Baltic Sea coastline [5]. Additionally, the enclosed characteristic of the Baltic Sea increases susceptibility to wind-induced sea level rise resulting in infrequent atmospheric pressure driven storm surges along the southern Baltic Sea coastline. In the past 60 years, the number of hours of storm surge with the entailing sea level rise of 0.5 m above sea level has increased significantly [6]. As a consequence, the adjacent terrestrial areas will be inundated by the Baltic Sea thereby facilitating hydrodynamic and biochemical exchange processes across the terrestrial–marine interface.

Nowadays, coastal terrestrial areas including farmlands and peatlands are often separated from the Baltic Sea by dunes and dikes as coastal protective measures. Due to this separation and the consequential artificial draining of the coastal areas, many peatlands have lost their original features and functions. Ecologically intact peatlands act as massive carbon sinks, that store up to twice as much carbon as all global forest biomass by covering only 3% of the world’s terrestrial area [7]. Therefore, the number of coastal peatlands being restored, including an active or passive removal of the coastal protective structures, is increasing [8].

Along the German Baltic Sea coastline, restoration of various coastal peatlands is in progress, including a costal fen within the nature reserve ‘Heiligensee and Hütelmoor’ in Mecklenburg Pomerania since the early 2000s [6]. The restoration plan includes removal of the near shore groyne systems as well as a gradual breakdown of the dune separating the Baltic Sea from this coastal peatland by natural forces, which resulted in a natural break in January 2019 due to an extremely strong storm surge (183 cm above mean sea level [9]). As a consequence, a massive Baltic Sea brackish water intrusion into the peatland took place. At the German Baltic Sea coast, additional peatlands are under restoration, such as the Drammendorfer Polder on the island Rügen, which was flooded by Baltic Sea water after a dike opening in November 2019 and a peatland in Karrendorf near Greifswald that has been under restoration for almost three decades [10,11]. These restoration projects strongly contribute to the storm surge-induced reinforced hydrodynamic and biogeochemical exchange processes across the sea–land transition zone, and also lead to the horizontal transportation of organisms and subsequently the intermixing of water bodies between peatlands and the coastal Baltic Sea.

Both ecosystems inhabit many microbial communities, including photosynthetic biofilms on top of soft bottom sediments, called microphytobenthos. The microphytobenthos is typically dominated by pennate diatoms that strongly contribute to the marine primary production [12,13]. Many pennate diatoms can vertically move in the sediment in response to environmental stressors [14,15] via the excretion of sticky extracellular polymeric substances (EPS) that stabilize soft bottom sediments [16]. Environmental stressors for these photosynthetic organisms mainly include light, temperature, and salinity gradients.

In both, coastal peatlands and the Baltic Sea, light availability seasonally and diurnally varies; however, both ecosystems also differ from each other in light conditions. Peatlands carry large amounts of dissolved and particulate organic matter resulting in high turbidity, which increase the incident light absorption [17]. This leads to a decrease in light penetration into the water column and hence low light conditions for benthic diatoms. In contrast, over the course of the seasons, the coastal shallow water of the Baltic Sea exhibits photon flux densities between 389 and 2117 µmol photons m^−2^ s^−1^, which saturate benthic diatom photosynthesis [18]. However, prevailing strong hydrodynamic forces in the Baltic Sea such as winds and currents can lead to the burial of benthic diatom cells, with at least a temporal lack of light or even darkness. As a response to such unfavorable light conditions, many benthic diatoms are capable to vertically move along light gradients [19].

Coastal peatland and the adjacent Baltic Sea also exhibit differences in the annual temperature despite the proximity to each other. Along the shallow southern Baltic Sea, temperature seasonally varies between 4 °C in winter and 22 °C in summer [20]; while in the peatlands, temperature ranges from around the freezing point in winter to around 30 °C in summer. The large content of humic substances in small peatland water bodies results in high absorption of solar radiation and hence enhanced increasing water temperatures in summer compared to the Baltic Sea. These distinguished habitat temperature conditions are reflected in the benthic diatom temperature requirements for photosynthesis since peatland isolates have displayed optima between 20 to 35 °C, whereas those from the Baltic Sea have exhibited highest oxygen production at 10 to 20 °C [17,21,22]. The different photo–physiological response patterns in peatland and Baltic Sea benthic diatoms can be biochemically explained as carbon fixation is mainly controlled by temperature-dependent enzymes [23].

Photosynthesis and growth of benthic diatoms is also strongly dependent on nutrient supply. As a result of their location on top of the sediment, benthic diatoms take up nutrients from two sources, e.g., the water column and the nutrient-rich pore water within the sediment [24]. Therefore, benthic diatoms are strongly involved in biogeochemical exchange processes such as vertical nutrient fluxes across the sediment water interface [25]. Besides inorganic compounds, peatlands are also rich in particulate and dissolved organic carbon due to low oxygen-related degradation of organic material [26]. Unlike most photosynthetic organisms, some benthic diatom species are known for a heterotrophic and/or mixotrophic lifestyle using organic compounds to fuel their metabolism and maintain their photosynthetic ability when deprived from light [27].

With the now increasing sea level and storm surges, restored peatlands are expected to experience stronger and more abundant inundation events leading to increased exchange and intermixing processes of the water bodies of peatland freshwater and the brackish Baltic Sea. However, the impact of such drastic environmental changes on benthic diatom growth and photosynthesis has never been investigated, despite the tremendous importance of these organisms for their habitats.

Therefore, two Baltic Sea and three peatland benthic diatom strains were investigated for their ecophysiological response patterns of growth, photosynthesis, and respiration under simulated intermixing processes, which included treatment with varying growth media, light, and temperature conditions. Due to higher amounts of nutrients and organic compounds in the peatland, we expected higher growth rates in peatland-based media. In addition, we assumed eurythermal traits and a high photo–physiological plasticity in all benthic diatoms.

## 2. Materials and Methods

### 2.1. Study Site

Along the German Baltic Sea coast, four study sites were sampled in 2019 using undisturbed sediment cores and Petri dishes (diameter: 5 cm) for the isolation of benthic diatom strains originating from the shallow Baltic Sea and adjacent peatland (Figure 1). Samples from the coastal Baltic Sea were isolated along the Baltic shoreline in front of the nature reserve “Heiligensee und Hütelmoor” (54.22550 N, 12.17185 E). In this area, abiotic parameters such as wind, temperature, and salinity strongly fluctuate and thereby constantly shape the entire sampling site. Peatland samples were obtained from benthal of three coastal peatlands no deeper then 20–30 cm below water surface. The second sampling station (54.21212 N, 12.18343 E) was located in the peatland within “Heiligensee und Hütelmoor” which is underlying sporadic exchange processes with the Baltic Sea in relation to the sea level. The peatland Drammendorf (54.36971 N, 13.24384 E) was sampled prior to a dike removal in 2019; therefore, samples were not affected by Baltic Sea water influence. The third peatland sample was obtained from Karrendorf (54.15796 N, 13.38859 E); due to its proximity to the Greifswalder Bodden had been irregularly flooded with Baltic Sea water for the past three decades. Generally, temperature of the Baltic Sea shallow water in front of the “Heiligensee und Hütelmoor” annually varied between 4 °C in winter and 22 °C in summer with salinities ranging around 8.2 to 15.3 S_A_ [22]. The three peatland sites showed a similar annual temperature range between 4 °C up to 30 °C, with the sampling site in Karrendorf falling dry in summer. Salinity typically fluctuated in these peatlands between 0.5 and 5 S_A_.

### 2.2. Culture Establishment and Culture Conditions

For the experimental set up, the previously identified four diatom strains with the respective GenBank IDs from Prelle et al. [22] were used: *Melosira nummuloides* (Melosiraceae, MW070612, strain PTM9a), *Planothidium* sp. (st. 1) (Achnanthidiaceae, MW070614, strain PTM25), *Planothidium* sp. (st. 2) (Achnanthidiaceae, MW070611, strain PTM7), and *Nitzschia filiformis* (Bacillariaceae, MW070613, strain PTM10). One additional strain originating from the Baltic Sea was used in the present study: *Hyalodiscus* cf. *scoticus* (Hyalodiscaceae, ON009273, strain PTM12). All strains were isolated according to Prelle et al. [22]. In short, the upper 1 cm sediment layer of each sediment core was used for the establishment of clonal cultures. Sediment was incubated for approximately two weeks in Guillard’s f/2 medium [28,29] enriched with metasilicate (Na_2_SiO_3_·5H_2_O; 10 g 100 mL^−1^) to a final concentration of 0.6 mM (further referred to as culture condition) using sterile filtered (0.45 µm) Baltic Sea water (approximately 12 S_A_) enriched with artificial sea salt (hw Marinemix^®^ professional, hw Wiegandt Aquaristik, Krefeld, Germany) to a final salinity of 15 S_A_ (further mentioned as culture cultivation media) at 20 °C at 30–50 μmol photons m^−2^ s^−1^ under a 16:8 h light:dark cycle (Osram Daylight Lumilux Cool White lamps L36W/840, Osram, Munich, Germany). Afterwards, single-cell isolation was performed using an inverted microscope (Olympus IX70, Olympus, Hamburg, Germany), until the establishment of unialgal cultures with low bacteria numbers ranging in maximum of 0.05–1% of the diatom volume (estimation via DAPI-staining). SEM (scanning electron microscopy) and light microscopy images were prepared following the same methodological approach as in Prelle et al. (2021), and are deposited in the Appendix A. Using molecular markers [22,30] and morphological identification with recent literature [31], diatom strains were determined. All cultures are available at the culture collection of the Department Applied Ecology and Phycology, University of Rostock. All diatom strains were kept in culture cultivation media at 20 °C, 30–50 μmol photons m^−2^ s^−1^ under a 16:8 h light:dark cycle.

### 2.3. Specific Growth Rates

Specific growth rates of the five diatom strains in response to 10 different growth media were investigated using in vivo fluorimetry following the approach of Karsten et al. [32], Gustavs et al. [33], and Prelle et al. [22]. This method uses chlorophyll, a fluorescence as a proxy for biomass and is particularly well suited for benthic diatoms.

The media were prepared with respect to possible flooding events between the Baltic Sea and adjacent peatland. These media were based on Baltic Sea water, peatland water, or a defined mixture of both (Table 1). All media were sterile filtered (0.45 µm) and stored at 5 °C until used. Salinity of the Baltic Sea water and the two saline peatland water growth media (GM9 and GM10) were adjusted to 15 S_A_ using artificial sea salt (hw Marinemix^®^ professional, hw Wiegandt Aquaristik, Krefeld, Germany). As a control, f/2 and metasilicate were added to the media GM1, GM8, and GM10. Additionally, 219 µM NaNO_3_, (Carl Roth, Karlsruhe, Germany) as a source of inorganic nitrogen and 27 µM NaH_2_PO_4_·H_2_O (Carl Roth, Karlsruhe, Germany) as a source of inorganic phosphorus were added to the peatland media (GM7, GM8, GM9, and GM10; proportional to the mixed media GM4, GM5, and GM6) to mimic the highest nutrient values found in the sampling site of “Heiligensee und Hütelmoor”.

All five diatom strains were cultivated in 15 mL of the respective medium (n = 3) in disposable Petri dishes with covered lids at 20 °C and 30–50 µmol photons m^−2^ s^−1^ under a 16:8 h light:dark cycle. Measurements proceeded every 24 h for 9 days by a MFMS fluorimeter (Hansatech Instruments, King’s Lynn, UK). Using blue LEDs (Nichia, Nürnberg, Germany) for the excitation of the chlorophyll *a* fluorescence; the resulting fluorescence was detected by an amplified photodiode and was separated from scattered excitation light through a long pass glass filter (RG 665; Schott, Mainz, Germany) and a bright-red gelatin filter (Lee, Brussels, Belgium) (see Karsten et al. [32] for further in-depth methodical details). The resulting relative fluorescence units were used for calculation of the specific growth rates (in µ d^−1^) during the logarithmic phase of the diatom strains applying the following equation: N = N_t_e^µdt^ (N—fluorescence intensity at the measuring day, N_t_—initial fluorescence intensity, dt—difference of time in days between measuring day and starting day, µ—growth rate) [33].

All five diatom strains were pre-incubated in the different media for four days before transfer to the experimental cultures to allow acclimation to the experimental condition.

### 2.4. Light Response Curves (PI-Curves)

The photosynthetic activity of the five diatom strains in response to increasing light availability and its dependency to the two selected cultivation media (GM6 and GM9) was measured in a so-called PI-Box following the approach of Prelle et al. [21]. These two media were chosen to investigate photosynthesis in likely occurring media changes after a storm surge events. The medium of GM2 was excluded due to a partially no growth response in the growth experiment. The PI-Box was constructed using water tempered oxygen electrode chambers (DW1, Hansatech Instruments, King’s Lynn, UK) and air-tight cover lids with incorporated oxygen dipping probe DP sensors (optodes) (PreSens Precision Sensing GmbH, Regensburg, Germany). These probes were connected to a fiber optic oxygen transmitter via optical fibers (Oxy 4mini meter, PreSens Precision Sensing GmbH, Regensburg, Germany), measuring oxygen within the samples using the concept of oxygen fluorescence quenching. Samples consisted of 3 mL log-phase algal suspension (n = 3) and the addition of 30 μL of sodium bicarbonate (NaHCO_3_, final concentration 2 mM), to avoid carbon deficit during measurements. Each sample was exposed to 10 increasing light levels that were between 0 and 1377 ± 44 μmol photons m^−2^ s^−1^ photosynthetically active radiation (PAR) via LEDs (LUXEON Rebel1 LXML-PWN1–0100, neutral-white, Phillips, Amsterdam, Netherlands) at 20 °C. Starting with a 30 min respirational phase in darkness, light levels were increased every 10 min until reaching maximum oxygen production. Even biomass distribution within the samples for the entire duration of the measurement was ensured by magnetic stirrers set under the chambers. After oxygen measurements, chlorophyll *a* of each sample was determined for normalization towards oxygen measurements using 96% ethanol (*v/v*) for extraction. Chlorophyll *a* was determined spectrophotometrically at 750 nm and 665 nm (Shimadzu UV-2401 PC, Kyoto, Japan) [34].

All data were fitted using the photosynthetic model of Walsby [35], and the accompanying PI-parameters net primary production (NPP_max_), respiration (R), light utilization coefficient (α), photoinhibition coefficient (β), light saturation point (I_k_), and the light compensation point (I_c_) were calculated.

### 2.5. Temperature Dependence of Photosynthesis and Respiration

The effect of increasing temperature between 5 and 40 °C in combination with two different cultivation media (GM6 and GM9) on photosynthetic and respiration rates of the five diatoms strains was determined following the approach of Karsten et al. [36] and Prelle et al. [21]. Using the same PI-Box as for the PI-curves, settings were adjusted to darkness during the respirational phase and photosynthesis saturated light conditions at 303.4 ± 11 µmol photons m^−2^ s^−1^ for the photosynthetic phase. Again, 3 mL of log-phase algal suspension and the addition of 30 μL of sodium bicarbonate were added to the oxygen electrode chambers as they were tempered at 5 °C. Measurements started with a 20 min dark incubation phase before measurements for the 10 min respirational and 10 min photosynthetic phase. Temperature was then increased by 5 °C and each step was repeated until reaching 40 °C. Afterwards, chlorophyll *a* was determined as was conducted for the PI-curves. All data were fitted using the photosynthetic model of Yan and Hunt [37], excluding photosynthesis at 40 °C for *H.* cf. *scoticus*.

### 2.6. Statistical Analysis

Statistical analysis followed the same approach as Prelle et al. [22]. Calculations, including solver function by minimizing the sum of normalized squared deviations, fitting of the photosynthetic model of Walsby [35] and image creation, were performed using Microsoft office Excel (Version: 2016, Microsoft Corporation, Redmond, WA, USA) and R (Version: 4.0.2, R Foundation for Statistical Computing, Vienna, Austria). Further, R was used for data fitting of the temperature-dependent photosynthesis using the model of Yan and Hunt [37], as well as calculation of significant levels using one-way ANOVA and the post-hoc Tukey’s significant differences test (critical *p*-value < 0.05). Significant differences were indicated by lowercase and capital letters. In the Appendix A, confidence intervals of the modeled data are shown, which were calculated using library nls tools in R.

## 3. Results

### 3.1. Species Identification

*Melosira nummuloides*, *Planothidium* sp. (st. 1 and 2) and *Nitzschia filiformis* were previously identified by Prelle et al. [22]. The additional strain was morphologically identified as *Hyalodiscus* cf. *scoticus* using light microscopy and SEM images. Further results for the molecular analysis of the *rbc*L gene using NCBI Blastn [38] corresponded to 99.7% with *Hyalodiscus scoticus* s0284 (AB430660) and to 99.7% with *Hyalodiscus* sp. CCMP1679 (FJ002131).

### 3.2. Growth

Two Baltic Sea (Figure 2A,B) and three peatland (Figure 2C–E) benthic diatom strains were investigated for their specific growth rate as a function of 10 different growth media. Overall, the peatland water-based media (GM7–GM10) resulted in similar specific growth rates for *H.* cf. *scoticus* (Figure 2B) and *N. filiformis* (Figure 2D), and significantly (*p* < 0.05) higher specific growth rates for *M. nummuloides* (Figure 2A), *Planothidium* sp. (st. 2) (Figure 2C) and *Planothidium* sp. (st. 1) (Figure 2E) compared to the standard cultivation media (GM1). Specific growth rates in GM1 ranged between 0.31 d^−1^ (*H.* cf. *scoticus*) and 0.64 d^−1^ (*Planothidium* sp. (st. 1)) (Figure 2B,E). In the peatland species, only small differences were determined among the peatland water-based media, from GM7 to GM10, displaying the overall highest specific growth rates between 0.67 and 0.76 d^−1^ (*Planothidium* sp. (st. 1)), 0.60 and 0.75 d^−1^ (*Planothidium* sp. (st. 2)), and 0.51 and 0.61 d^−1^ (*N. filiformis*) (Figure 2C,D). The two Baltic Sea isolates *M. nummuloides* and *H.* cf. *scoticus* exhibited specific growth rates similar to those in the cultivation media for the saline peatland media (GM9 and GM10), between 0.56 and 0.67 d^−1^ and 0.27 and 0.38 d^−1^, respectively (Figure 2A,B). A general trend for all five strains can be depicted for GM4, GM5, and GM6, as with the increasing proportion of peatland water-specific growth rates were stimulated by at least two times, although still slightly lower compared to GM7–GM10 (Figure 2). GM2 resulted in the overall lowest specific growth rates in all strains ranging between no growth and 0.12 d^−1^, respectively. The addition of nutrients (NaNO_3_ and NaH_2_PO_4_·H_2_O) to the Baltic Sea water (GM3) had a significant effect on specific growth rates of *M. nummuloides* and both *Planothidium* sp. strains compared to GM2 (Figure 2A,C,E). However, these rates were at least two thirds lower than highest overall specific growth rates under GM1, which also contained Na_2_SiO_3_ in addition to NaNO_3_ and NaH_2_PO_4_·H_2_O.

### 3.3. PI-Curves

The light curves in combination with two growth media, GM6 and GM9, were investigated for both Baltic Sea (Figure 3A–D) and all peatland (Figure 4A–F) diatom strains, and the resulting parameters are shown in Table 2. The overall highest NPP_max_ was measured in *Planothidium* sp. (st. 2) with 79.6 ± 19.5 µmol O_2_ mg^−1^ chl *a* h^−1^ (GM9) and significant differences between treatments (*p* < 0.05, Table 2). Highest respiration was estimated with −79.5 ± 21 (*M. nummuloides*, GM6) and lowest with −23.9 ± 2.3 µmol O_2_ mg^−1^ chl *a* h^−1^ (*Planothidium* sp. (st. 1), GM9), and varied significantly (*p* < 0.05) between species (Table 2). However, no significant differences between media were determined. Light compensation points (I_c_) ranged between 14.2 ± 2 µmol photons m^−2^ s^−1^ (*Planothidium* sp. (st. 1), GM9) and 46.2 ± 25.7 (*M. nummuloides*, GM6), while light saturation points (I_k_) were higher between 19.6 ± 7.3 (*H.* cf. *scoticus*, GM6) and 68.0 ± 7 (*Planothidium* sp. (st. 2), GM9) (Table 2). While photoinhibition (β) was found in half of the strains, but with relatively low values between −0.01 and −0.02 µmol O_2_ mg^−1^ chl *a* h^−1^ (µmol photons m^−2^ s^−1^)^−1^, no photoinhibition was detected in the remaining samples (Table 2).

### 3.4. Effect of Temperature on Photosynthesis and Respiration

The temperature increase from 5 °C to 40 °C in all benthic diatom strains led to a similar response pattern of the photosynthetic and respirational rates with significantly (*p* < 0.05) increasing signals until reaching an optimum followed by a strong decline (Figure 5A–D and Figure 6A–F). Except for *H.* cf. *scoticus*, both cultivation media compared to each other did not influence the photosynthesis/respiration rates under rising temperatures (Figure 5C,D). *Hyalodiscus* cf. *scoticus* exhibited a broader temperature tolerance for photosynthesis in GM9. Optimum temperature for photosynthesis of the Baltic Sea species was between 5 and 30 °C compared to 10 and 35 °C in the peatland species, with a similar pattern also observed for respiration (Figure 7 and Figure 8). The photosynthetic rates of 20–80% of the maximum were between 5 and 35 °C for all three peatland strains (Figure 8); while in the Baltic Sea strains, only 0–19% of the maximum was reached at 35 °C (Figure 7). Generally, both cultivation media led to a shift in optimum ranges of the photosynthetic rates, with the most pronounced differences in *N. filiformis* and *H*. cf. *scoticus*. While treatment with GM9 led to a lower optimum temperature (10–20 °C) compared to GM6 (20–30 °C) in *N. filiformis*, *H.* cf. *scoticus* displayed a reverse pattern with lower optimum temperature for GM6 at 5 °C compared to GM6 at 25 °C (Figure 7A and Figure 8A). A similar pattern could not be identified in the respirational rates of both strains (Figure 7A and Figure 8A). Fitting of the data sets using the model of Yan and Hunt [37] also generally resulted in lower temperature requirements for optimum photosynthesis for the Baltic Sea strains compared to the peatland isolates, while temperature requirements for optimum respiration was rather similar (Table 3 and Table 4). Treatment with both media resulted in higher maximum photosynthetic rates and lower respiration rates for all peatland strains and *H*. cf. *scoticus* in GM9 compared to GM6 (Table 3 and Table 4); while reversely, GM6 stimulated higher photosynthetic and respirational rates in *M. nummuloides* (Table 3).

## 4. Discussion

As already mentioned in the introduction, with the increasing sea level rise and strength of storm surges along the southern Baltic Sea, coastal peatlands will be increasingly more affected by brackish water inundation and the resulting biota transported in the water masses, with so far unstudied ecological consequences for such wetlands. Therefore, we simulated different habitat conditions and evaluated the effects on ecophysiological traits of benthic diatoms as important primary producers, which originated from the shallow Baltic Sea and adjacent coastal peatlands.

### 4.1. Growth

The flooding of coastal peatlands by Baltic Sea water results in the establishment of different physico–chemical conditions for the inhabiting of aquatic organisms. With respect to such processes, 10 different experimental growth media consisting of different proportions of Baltic Sea and peatland water and different nutrient conditions resulted in species-specific responses with significantly higher specific growth rates of up to 0.8 d^−1^ in the peatland-based media. From an ecological perspective, growth is the physiological key indicator of the organism’s fitness in response to its environment, which is controlled by multiple abiotic factors such as light conditions and temperature in benthic diatoms [39,40,41]. Growth is also driven by the availability of inorganic nutrients to fuel the photosynthetic metabolism [42]. Typically, coastal fens carry large amounts of the essential macronutrients nitrogen and phosphorus, due to low oxygen levels and the resulting low redox potentials. Therefore, benthic diatoms benefit from the high nutrient availability in the peatland-based media as reflected in the high specific growth rates of both peatland and Baltic Sea originating strains.

The two Baltic Sea strains *Melosira nummuloides* and *Hyalodiscus* cf. *scoticus* did not grow in freshwater-based peatland media compared to the saline peatland media, pointing to some salt requirements. Salinity is known to influence growth on a species-specific basis [43,44], and *M. nummuloides*, for example, is described from marine habitats [45]. Corresponding to the present study, Prelle et al. [22] reported a euryhaline growth response for *M. nummuloides* from 5 to 39 S_A_, with inhibited growth at 1 S_A_. In addition, *Planothidium* sp. (st. 2), *Nitzschia filiformis* and *Planothidium* sp. (st. 1) exhibited euryhaline growth responses between 1 and 39 S_A_ [22], supporting the present findings of growth in all tested peatland water-based media.

The intermixing process between the Baltic Sea and coastal peatlands would be a rather transitional process, resulting in different proportions of peatland to Baltic Sea water masses. Under such conditions, all five benthic diatom strains, regardless of their originating habitat, were in principle able to exhibit growth, reflecting at least an initial growth phase after a storm surge-induced intermixing of water masses from both habitats. However, the limited lower salinity tolerance of the Baltic Sea strains will not allow permanent occurrence in the peatland with further re-freshening of the wetland.

On the other hand, peatland diatoms transferred to the Baltic Sea medium did not grow, and the same was true for the Baltic Sea strains, which can be explained by nutrient deficiency. Measurements for nitrogen and phosphorus in the Baltic Sea medium compared to the modified peatland water-based media revealed much lower concentrations, which although reflecting the monthly nutrient measurements for the shallow Baltic Sea water column ranging between 1.2 and 11.2 µmol/l for nitrogen (as NO_3_^−^, NO_2_^−^ and NH_4_^+^ combined) and 0.1 to 1.2 µmol/L for dissolved and hence bioavailable orthophosphate (Prelle, unpublished data), they do not reflect sediment pore water conditions. Low availability of nitrogen and phosphorus can lead to nutrient deficiency in microalgae, thereby lowering specific growth rates [46]. Under natural conditions, benthic diatoms typically benefit from the nutrient-rich pore water [24]. Nevertheless, treatment of the nitrogen and phosphorous enriched Baltic Sea medium (GW3) did not lead to enhanced specific growth rates as found for the peatland media, indicating other compounds (e.g., trace metals, vitamins), which might be essential for diatom growth.

Compared to other phototrophic organisms, silicate is one of the key nutrients for diatom growth, as it is used for the biosynthesis of the characteristic silicate frustule [47]. In contrast to the peatland water, the Baltic Sea showed relatively low silicate concentrations (Baltic Sea < 1.0 SiO_2_ mg L^−1^, peatland > 6.0 SiO_2_ mg L^−1^; using the SiO_2_ visual aquarium quick assay (JBL, Neuhofen Germany)). Therefore, experimental strains may have suffered silicate deficiency as reflected in low or declined growth. Recent studies conducted in the Baltic Sea in front of the Hütelmoor reported submarine ground water discharge from the peatland into the Baltic Sea [48]. Further, nutrient fluxes between the pore water and the overlaying water column were increased up to four times under typical hydrodynamic conditions found in this area. In addition, pore water was analyzed using oxygen isotopes proving a significant source of nitrogen, phosphorus, and silicate [6]. Therefore, benthic diatoms should be supplied with sufficient silicate from the pore water when inhabiting the sediment surface.

As already mentioned, growth was additionally stimulated by the peatland-derived medium. Due to low decomposition rates as a result of low oxygen levels, peatland waters carrying high amounts of dissolved organic carbon and across the sediment–water interface of the Baltic Sea in front of the Hütelmoor, around 250 mg/L were measured [6] compared to 1.1–4.8 mg/L in the water column at different sites [49]. The available organic carbon could have fueled a heterotrophic and/or mixotrophic metabolism in the diatoms. Both facultative and obligate heterotrophic pathways have been reported in diatoms [27] using organic compounds such as glucose via glycolysis to fuel their metabolism [50]. While the chemical composition of organic compounds in the utilized peatland water is unclear, the study by Prelle et al. [21] indicated heterotrophic growth in the Baltic Sea diatom *Actinocyclus octonarius* incubated in Hütelmoor peatland water. So far, detailed studies on heterotrophic or mixotrophic growth in the Baltic Sea and peatland benthic diatoms are missing.

Intermixing of water masses from both habitats may also lead to changes in pH, as the Baltic Sea exhibits higher pH values at 8.3 than the peatland at 6.8 (Prelle and Mutinova, unpublished data). At least the planktonic diatom *Skeletonema costatum* was not affected in growth by pH of 6.5 and 8.5 [51], and we assume similar response for benthic diatoms.

### 4.2. Light

The strong turbidity of peatlands results in lower light conditions in the same depths compared to the Baltic Sea, pointing to benthic diatoms in wetlands with lower light requirements than those in the Baltic Sea. Calculated PI-curve parameters confirm this assumption, as maximum photosynthetic rate was already reached between 25 and 68 µmol photons m^−2^ s^−1^ in combination with low light compensation points ranging between 19 and 32 µmol photons m^−2^ s^−1^ in the three peatland diatom strains. The Baltic Sea benthic diatom strains exhibited light saturated photosynthesis between 20 and 61 µmol photons m^−2^ s^−1^ but slightly higher light compensation points of 27–46 µmol photons m^−2^ s^−1^ indicating low light acclimation. Despite stronger light penetration in the Baltic Sea compared to the peatland, the shallow coastal Baltic Sea is a dynamic wind-driven environment leading to strong sediment resuspension, thereby causing regularly low light conditions due to turbidity or even cell burial. On the other hand, transportation into shallower depths within both habitats can lead to the exposure to higher photon fluence rates. All five diatom strains exhibited unaffected photosynthesis over wide photon fluence rates with only minor photoinhibition at up to 1400 µmol photons m^−2^ s^−1^, indicating a high photo–physiological tolerance and plasticity. In case of unfavorable high or low light conditions, diatoms reveal an array of acclimation mechanisms. On top of the sediment, pennate diatoms avoid high light exposure by vertically moving down into the sediment via the excretion of EPS, such as found in *N. filiformis* [14,15,52]. Diatoms have additional biochemical abilities to avoid oxidative damage of the photosystem II via the dissipation of excess energy as heat resulting from overexposure to strong light conditions [53]. The process of non-photochemical quenching of chlorophyll *a* fluorescence is induced within a few seconds to minutes [53]. Further, an accumulation of the photoprotective xanthophyll pigment diatoxanthin via the reaction of diadinoxanthin de-epoxidation weakens the oxidative stress under high light conditions [54]. However, other mechanisms such as the antioxidative potential (antioxidants and antioxidative enzymes) [55] must be considered for light acclimation. Photosynthetic activity of the five diatom strains was not affected by the different media.

### 4.3. Temperature

Contrary to the very dynamic coastal Baltic Sea, the adjacent peatlands are characterized as shallow, still, and often dark habitats with a low albedo. In summer, these traits lead to a rapid warming of the respective water bodies reaching over 30 °C, while in winter temperatures around the freezing point and ice coverage can be observed (personal observation).

Generally, all five diatoms exhibited broad temperature tolerances with moderate photosynthesis from 5 to 35 °C. Despite the differences in annual temperature ranges of the respective habitats and species-specific responses, rather similar photosynthetic rates were found in the peatland and Baltic Sea diatoms, independent of the media tested. In accordance with other studies, the optimum temperature for photosynthesis is generally found to be lower compared to the respiration [21,56,57]. The obvious decoupling of both processes may be explained by the photosynthetic light reactions mainly being light driven, and even indicated to be rather unaffected by temperature [58] in comparison to respiration, which is mainly controlled by temperature-dependent enzymes [59,60]. Nevertheless, in the photosynthetic dark reaction the carbon fixing enzyme RuBisCO (Ribulose-1,5-bisphosphat carboxylase-oxygenase) is losing its specificity towards CO_2_ with increasing temperature [61].

## 5. Conclusions

In conclusion, all five benthic diatom strains exhibited high photo–physiological plasticity and eurythermal traits that would allow for survival in both the Baltic Sea and the peatlands. A storm surge driven intermixing process of water masses from both ecosystems would generally facilitate diatom growth, but only in the presence of sufficient salinity for Baltic Sea isolates. More importantly, our data clearly show that peatland water enhances growth with the indication of heterotrophic growth. Overall, benthic diatoms from the Baltic Sea and the adjacent peatlands are very well acclimated to respond to the described intermixing processes along the terrestrial–marine ecocline resulting from sea level rise and storm surges.

## Figures and Tables

**Figure 1 microorganisms-10-00749-f001:**
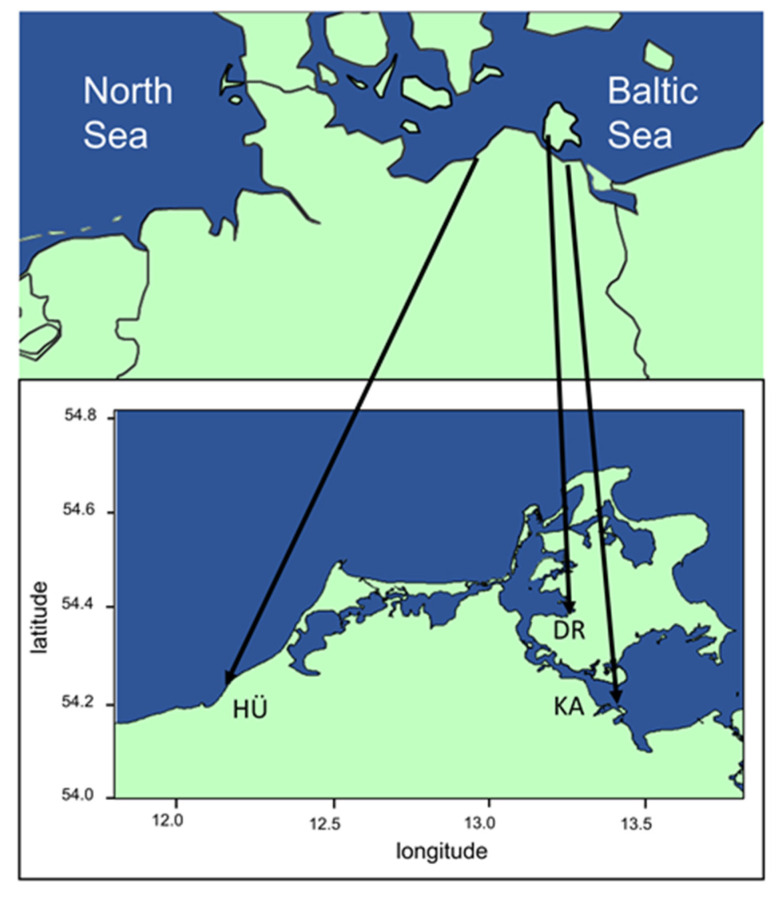
Map of northern Germany showing the three sampling sites HÜ (Hütelmoor), DR (Drammendorf) and KA (Karrendorf) next to the southern Baltic Sea.

**Figure 2 microorganisms-10-00749-f002:**
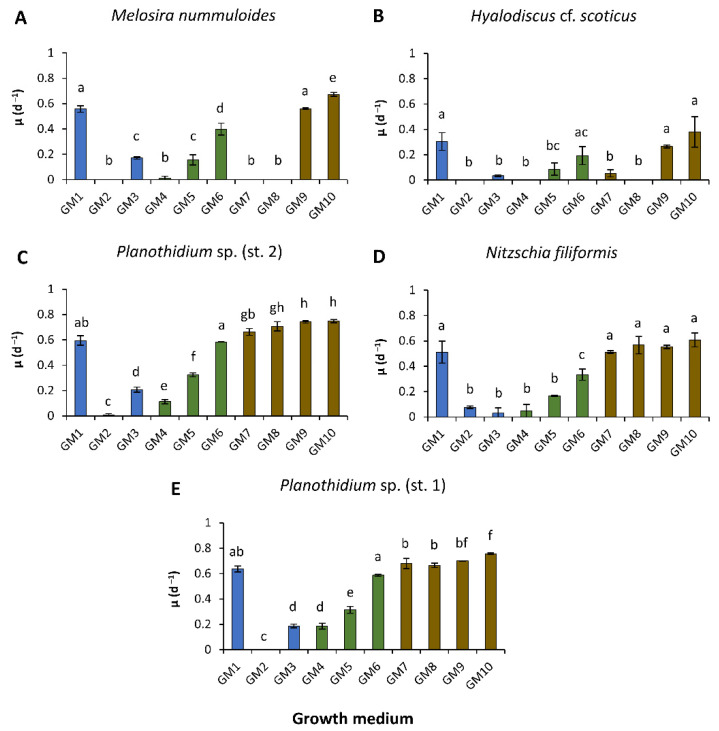
Specific growth rates (d^−1^) in relation to 10 growth media (GM) (blue = Baltic Sea water base, brown = peatland water base, green = mixed water base) of two Baltic Sea (**A**,**B**) and three peatland benthic diatom (**C**–**E**) strains. Data represent mean values ± standard deviation (n = 3). Different lowercase letters represent significant levels among all means as calculated by a one-way ANOVA (Tukey’s test, *p* < 0.05). (**A**) *Melosira nummuloides*, (**B**) *Hyalodiscus* cf. *scoticus*, (**C**) *Planothidium* sp. (st. 2), (**D**) *Nitzschia filiformis,* and (**E**) *Planothidium* sp. (st. 1).

**Figure 3 microorganisms-10-00749-f003:**
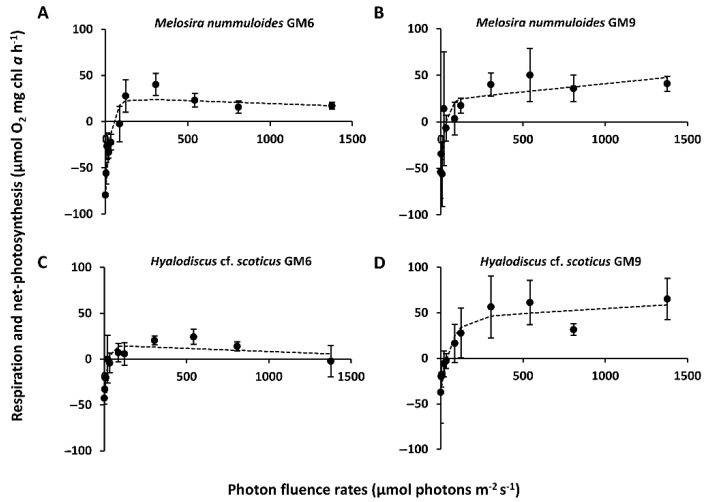
Photosynthesis and respiration rates (μmol O_2_ mg^−1^ chl *a* h^−1^) of light curves (μmol photons m^−2^ s^−1^) for two Baltic Sea benthic diatom strains kept at 20 °C in GM6 medium (**A**,**C**) and GM9 medium (**B**,**D**). Data represent mean values ± standard deviation (n = 3, except *Melosira nummuloides* GM6 n = 4, *Hyalodiscus* cf. *scoticus* GM6 n = 2). (**A**) *Melosira nummuloides* GM6, (**B**) *Melosira nummuloides* GM9, (**C**) *Hyalodiscus* cf. *scoticus* GM6, and (**D**) *Hyalodiscus* cf. *scoticus* GM9. Data points were fitted by the model of Walsby [35].

**Figure 4 microorganisms-10-00749-f004:**
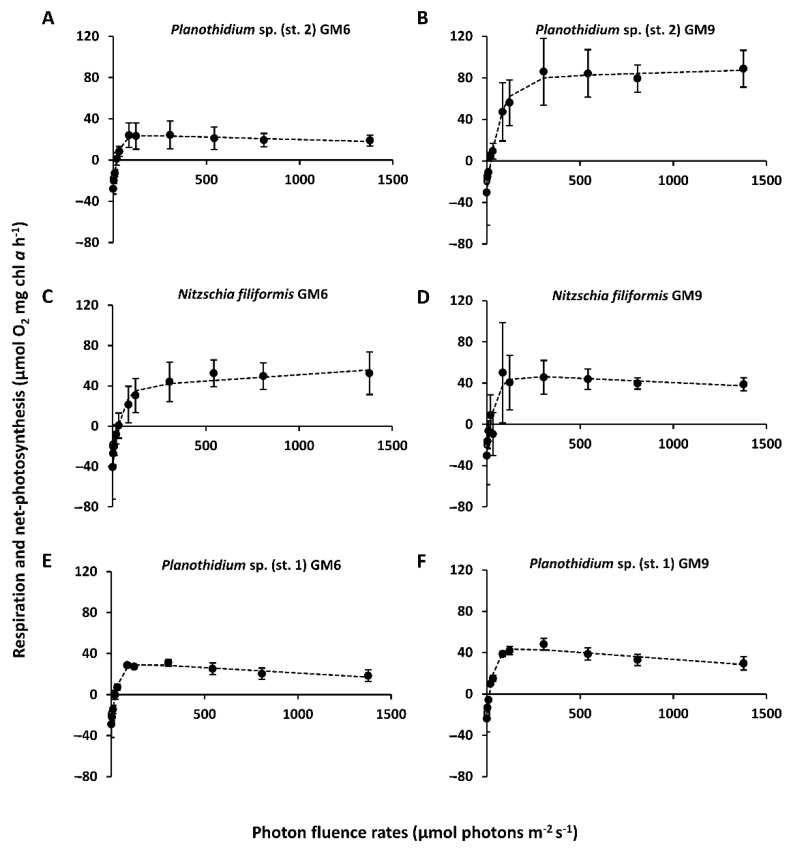
Photosynthesis and respiration rates (μmol O_2_ mg^−1^ chl *a* h^−1^) of light curves (μmol photons m^−2^ s^−1^) for three peatland benthic diatom strains kept at 20 °C in GM6 medium (**A**,**C**,**E**) and GM9 medium (**B**,**D**,**F**). Data represent mean values ± standard deviation (n = 3, except *Planothidium* sp. (st. 2) GM6 and *Nitzschia filiformis* GM6 n = 4,). (**A**) *Planothidium* sp. (st. 2) GM6, (**B**) *Planothidium* sp. (st. 2) GM9, (**C**) *Nitzschia filiformis* GM6, (**D**) *Nitzschia filiformis* GM9, (**E**) *Planothidium* sp. (st. 1) GM6, and (**F**) *Planothidium* sp. (st. 1) GM9. Data points were fitted by the model of Walsby [35].

**Figure 5 microorganisms-10-00749-f005:**
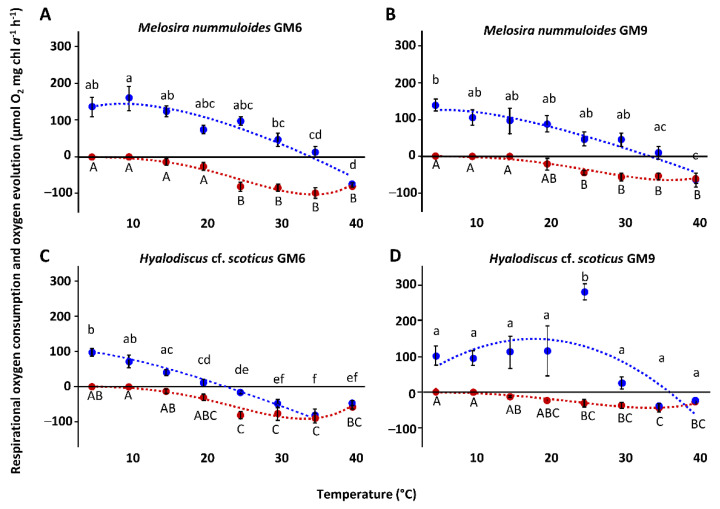
Photosynthetic (blue) oxygen production at 303.4 ± 11 µmol photons m^−2^ s^−1^ and respiratory (red) oxygen consumption in darkness of two Baltic Sea benthic diatom strains as a function of increasing temperature. The measured data were fitted by the model of Yan and Hunt [37] (photosynthesis: blue dashed line (**C**), 40 °C excluded); respiration: red dashed line). Cultures were kept in GM6 media (**A**,**C**) and GM9 media (**B**,**D**). Data represent mean values ± standard deviation (n = 3). Different lowercase (photosynthesis) and capital letters (respiration) indicate significant means (*p* < 0.05, one-way ANOVA with post hoc Tukey’s test). (**A**) *Melosira nummuloides* GM6, (**B**) *Melosira nummuloides* GM9, (**C**) *Hyalodiscus* cf. *scoticus* GM6, and (**D**) *Hyalodiscus* cf. *scoticus* GM9.

**Figure 6 microorganisms-10-00749-f006:**
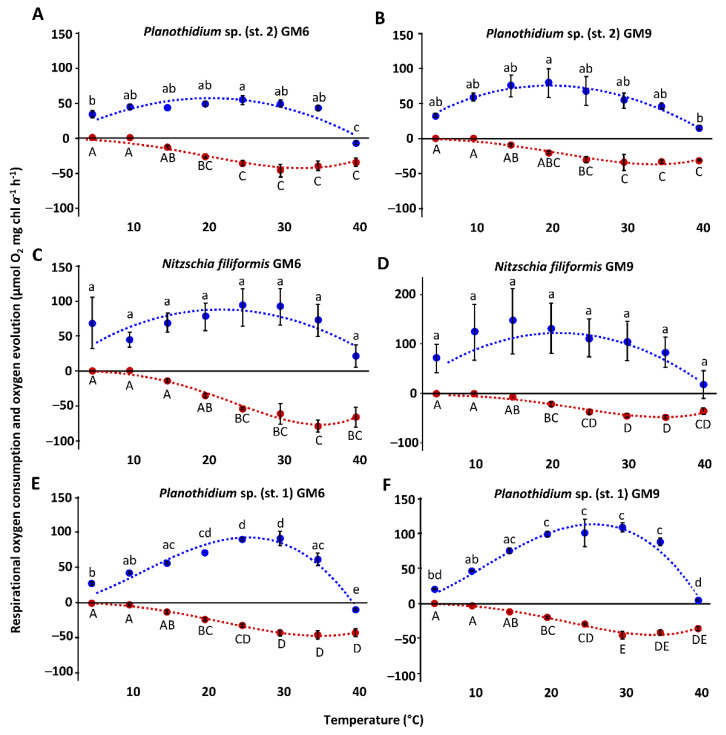
Photosynthetic (blue) oxygen production at 303.4 ± 11 µmol photons m^−2^ s^−1^ and respiratory (red) oxygen consumption in darkness of three peatland benthic diatom strains as a function of increasing temperature. The measured data were fitted by the model of Yan and Hunt [37] (photosynthesis: blue dashed line; respiration: red dashed line). Cultures were kept in GM6 media (**A**,**C**,**E**) and GM9 media (**B**,**D**,**F**). Data represent mean values ± standard deviation (n = 3). Different lowercase (photosynthesis) and capital letters (respiration) indicate significant means (*p* < 0.05, one-way ANOVA with post hoc Tukey’s test). (**A**) *Planothidium* sp. (st. 2) GM6, (**B**) *Planothidium* sp. (st. 2) GM9, (**C**) *Nitzschia filiformis* GM6, (**D**) *Nitzschia filiformis* GM9, (**E**) *Planothidium* sp. (st. 1) GM6, and (**F**) *Planothidium* sp. (st. 1) GM9.

**Figure 7 microorganisms-10-00749-f007:**
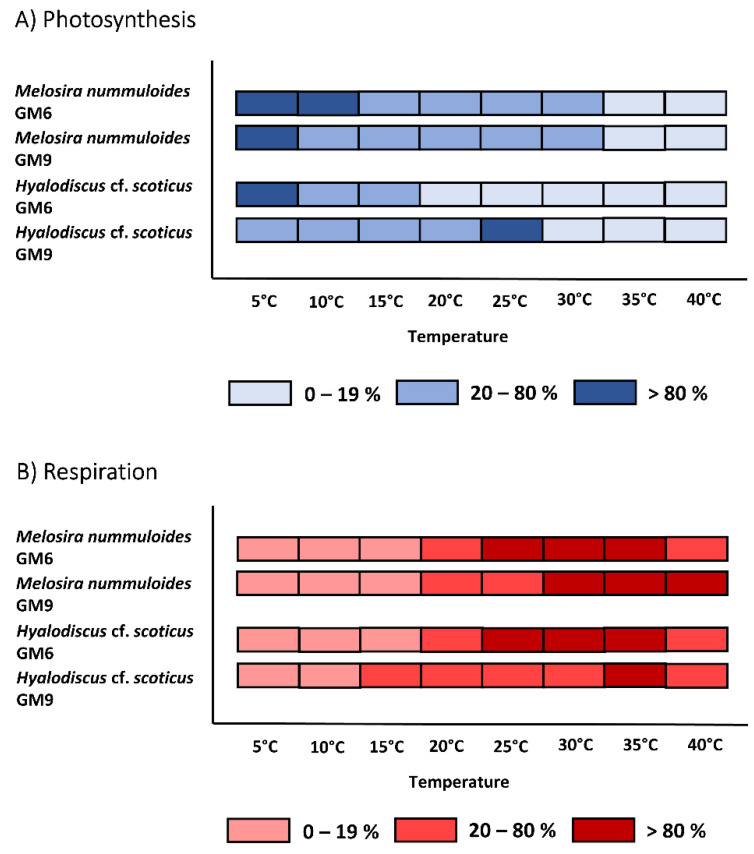
Effect of temperature on (**A**) photosynthetic oxygen evolution and (**B**) respirational oxygen consumption of two Baltic Sea benthic diatom strains in two different cultivation media (GM6 and GM9). Dark blue/red symbols represent highest photosynthesis at >80% percentile, medium blue/red symbols within 20 and 80% percentile and light blue/red symbols < 20% percentile. Data represent mean values (n = 3).

**Figure 8 microorganisms-10-00749-f008:**
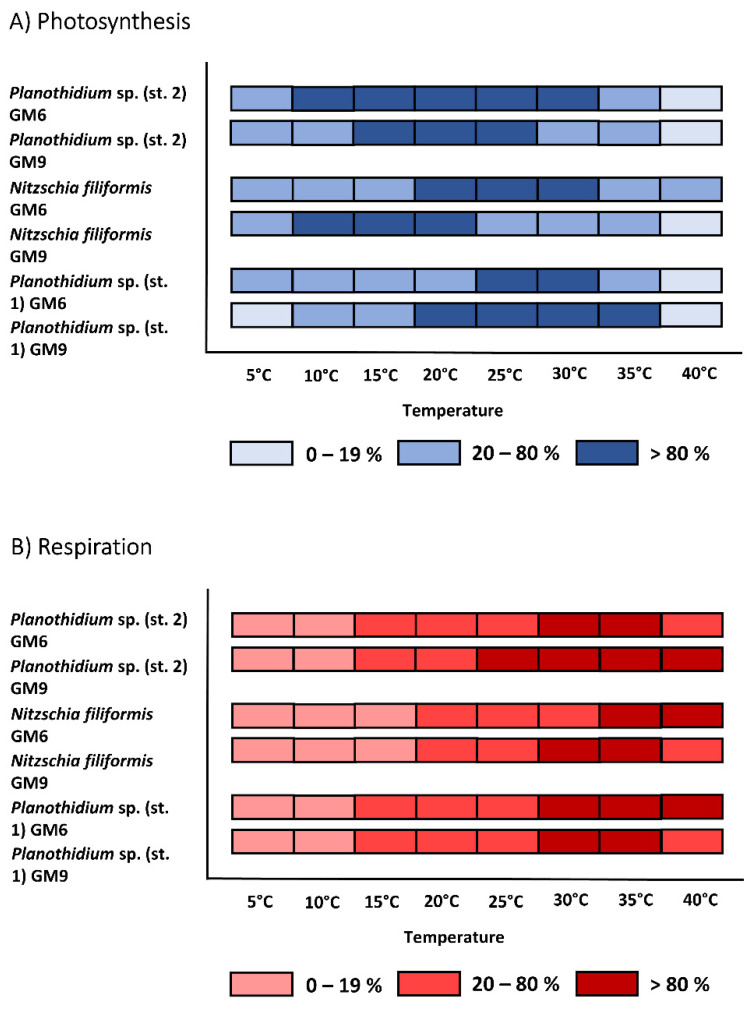
Effect of temperature on (**A**) photosynthetic oxygen evolution and (**B**) respirational oxygen consumption of three peatland benthic diatom strains in two different cultivation media (GM6 and GM9). Dark blue/red symbols represent highest photosynthesis at >80% percentile, medium blue/red symbols within 20 and 80% percentile and light blue/red symbols < 20% percentile. Data represent mean values (n = 3).

**Table 1 microorganisms-10-00749-t001:** Media composition for experimental growth medium GM1 to GM10 with the respective salinities (S_A_) and respective nutrient additions of f/2 + meatsilicate, inorganic nitrogen (N) as NaNO_3_ and inorganic phosphorous (P) as NaH_2_PO_4_·H_2_O to supplement water basis.

Medium	Water Base	Salinity (S_A_)	Nutrient Addition	pH
**GM1**	Baltic Sea	15	f/2 + metasilicate	7.6
**GM2**	Baltic Sea	15	-	7.4
**GM3**	Baltic Sea	15	219 µM N + 27 µM P	7.9
**GM4**	49/50 Baltic Sea + 1/50 peatland	15	219 µM N + 27 µM P (in the proportional peatland water)	7.6
**GM5**	9/10 Baltic Sea + 1/10 peatland	14.5	219 µM N + 27 µM P (in the proportional peatland water)	7.4
**GM6**	½ Baltic Sea + ½ Peatland	8	219 µM N + 27 µM P (in the proportional peatland water)	7.2
**GM7**	Peatland	0.3	219 µM N + 27 µM P (in the proportional peatland water)	7.8
**GM8**	Peatland	0.3	f/2 + metasilicate	7.4
**GM9**	Peatland	15	219 µM N + 27 µM P (in the proportional peatland water)	7.1
**GM10**	Peatland	15	f/2 + metasilicate	7.3

**Table 2 microorganisms-10-00749-t002:** Parameter of respective PI-curves (Figure 3A–D and Figure 4A–F) for five benthic diatom strains (n = 3) kept at 20 °C in GM6 and GM9 media. Data represent mean values ± standard deviation (n = 3, except *Melosira nummuloides* GM6 n = 4, *Hyalodiscus* cf. *scoticus* GM6 n = 2). Different lowercase letters represent significant levels among all means as calculated by a one-way ANOVA (Tukey’s test, *p* < 0.05). NPP_max_ represents the maximal oxygen production rate, alpha the initial slope of production in the light limited range, beta the terminal slope of production in extensive light range (photoinhibition), I_k_ the light saturation point, and I_c_ the light compensation point.

Isolates	NPP_max_ (µmol O_2_ mg^−1^ chl *a* h^−1^)	Respiration (µmol O_2_ mg^−1^ chl *a* h^−1^)	α (µmol O_2_ mg^−1^ chl *a* h^−1^) (µmol Photons m^−2^ s^−1^)^−1^	β (µmol O_2_ mg^−1^ chl *a* h^−1^) (µmol Photons m^−2^ s^−1^)^−1^	I_k_ (µmol Photons m^−2^ s^−1^)	I_c_ (µmol Photons m^−2^ s^−1^)	NPP_max_: Respiration
** *Melosira nummuloides GM6* **	23.31 ± 14.55 *^a^*	−79.45 ± 21.32 *^cd^*	3.23 ± 2.57 *^a^*	−0.1 ± 0.01 *^a^*	32.13 ± 21.40 *^ac^*	46.19 ± 25.69 *^a^*	0.31 ± 0.12 *^d^*
** *Melosira nummuloides GM9* **	23.47 ± 18.07 *^a^*	−53.90 ± 4.81 *^bd^*	2.78 ± 2.38 *^a^*	0.02 ± 0.01 *^a^*	27.86 ± 22.03 *^abc^*	32.58 ± 19.60 *^a^*	0.44 ± 0.32 *^acd^*
***Hyalodiscus* cf. *scoticus GM6***	14.12 ± 11.22 *^a^*	−42.62 ± 4.66 *^ab^*	2.89 ± 1.26 *^a^*	−0.01 ± 0.02 *^a^*	19.61 ± 7.33 *^a^*	26.99 ± 26.75 *^a^*	0.33 ± 0.30 *^cd^*
***Hyalodiscus* cf. *scoticus GM9***	43.75 ± 27.66 *^a^*	−37.29 ± 5.83 *^ab^*	1.32 ± 0.81 *^a^*	0.01 ± 0.01 *^a^*	61.39 ± 16.49 *^abc^*	37.28 ± 22.63 *^a^*	1.17 ± 0.54 *^acd^*
** *Planothidium sp. (st. 2) GM6* **	23.88 ± 13.26 *^a^*	−27.89 ± 4.70 *^ab^*	2.07 ± 0.27 *^a^*	0.00 ± 0.1 *^a^*	25.07 ± 8.18 *^a^*	19.33 ± 1.94 *^a^*	0.86 ± 0.34 *^acd^*
** *Planothidium sp. (st. 2) GM9* **	79.58 ± 19.51 *^b^*	−30.48 ± 0.27 *^ab^*	1.62 ± 0.47 *^a^*	0.01 ± 0.00 *^a^*	68.03 ± 7.06 *^abc^*	21.95 ± 5.58 *^a^*	2.61 ± 0.60 *^e^*
** *Nitzschia filiformis GM6* **	38.54 ± 8.42 *^a^*	−40.68 ± 11.32 *^ab^*	1.78 ± 1.22 *^a^*	0.01 ± 0.01 *^a^*	44.50 ± 40.43 *^abc^*	31.61 ± 26.22 *^a^*	0.95 ± 022 *^acd^*
** *Nitzschia filiformis GM9* **	46.62 ± 23.12 *^ab^*	−30.32 ± 8.12 *^ab^*	1.89 ± 1.12 *^a^*	−0.01 ± 0.02 *^a^*	40.63 ± 9.98 *^abc^*	20.33 ± 13.04 *^a^*	1.54 ± 0.51 *^ab^*
** *Planothidium sp. (st. 1) GM6* **	29.60 ± 1.99 *^a^*	−28.95 ± 2.79 *^ab^*	1.88 ± 0.42 *^a^*	−0.01 ± 0.00 *^a^*	31.07 ± 7.68 *^ac^*	21.10 ± 4.13 *^a^*	1.02 ± 0.14 *^acd^*
** *Planothidium sp. (st. 1) GM9* **	44.01 ± 3.90 *^a^*	−23.86 ± 2.34 *^ab^*	2.07 ± 0.73 *^a^*	−0.01 ± 0.00 *^a^*	32.77 ± 6.05 *^a^*	14.19 ± 1.97 *^a^*	1.84 ± 0.20 *^abc^*

**Table 3 microorganisms-10-00749-t003:** Results of model calculation for temperature-dependent photosynthetic and respirational rate (µmol O_2_ mg^−1^ chl *a* h^−1^) of two Baltic Sea benthic diatom strains in the growth media GM6 and GM9 (Table 1) following the model of Yan and Hunt [37].

			*Melosira nummuloides* GM6	*Melosira nummuloides* GM9	*Hyalodiscus* cf. *scoticus* GM6	*Hyalodiscus* cf. *scoticus* GM9
**Photosynthesis**	maximal photosynthetic rate		144.5	125.9	74.1	148.8
	optimum temperature (°C)		9.0	6.3	4.2	18.2
	maximum temperature (°C)		34.4	33.8	22.5	36.4
	residual sum-square		24,794	28,054	7083	158,658
	temperature (°C) range for	optimal photosynthesis (80% photosynthetic rate)	2.5–18.8	1.1–16.1	0.7–10.6	10.1–26.4
		photosynthesis (20% photosynthetic rate)	0.0–31.3	0.0–29.9	0.0–19.9	1.9–34.5
**Respiration**	maximal respirational rate		−100.9	−64.5	−90.6	−42.5
	optimum temperature (°C)		34.6	36.6	33.3	33.7
	maximum temperature (°C)		44.6	47.7	43.3	45.0
	residual sum-square		5415	4764	7797	1704
	Temperature (°C) range for	optimal respiration (80% respirational rate)	28.1–39.6	29.4–42.1	26.8–38.3	26.6–39.3
		respiration (20% respirational rate)	16.4–45.8	16.4–46.7	15.0–42.4	14.1–43.9

**Table 4 microorganisms-10-00749-t004:** Results of model calculation for temperature-dependent photosynthetic and respirational rate (µmol O_2_ mg^−1^ chl *a* h^−1^) of three peatland benthic diatom strains in the growth media GM6 and GM9 (Table 1) following the model of Yan and Hunt [37].

			*Planothidium sp. (st. 2)* GM6	*Planothidium sp. (st. 2)* GM9	*Nitzschia filiformis* GM6	*Nitzschia filiformis* GM9	*Planothidium sp. (st. 1)* GM6	*Planothidium sp. (st. 1)* GM9
**Photosynthesis**	maximal photosynthetic rate		57.1	76.1	87.9	138.6	93.1	113.3
	optimum temperature (°C)		20.6	20.1	22.1	18.4	25.7	26.1
	maximum temperature (°C)		40.7	42.4	45.3	42.2	39.7	40.7
	residual sum-square		2699	7837	30,816	95,423	2833	4240
	temperature (°C) range for	optimal photosynthesis (80% photosynthetic rate)	11.5–29.6	10.6–29.9	11.9–32.4	8.9–28.7	17.9–32.4	18.0–33.0
		photosynthesis (20% photosynthetic rate)	2.3–38.6	1.7–40.0	2.1–42.9	1.1–39.5	6.7–38.4	6.6–39.3
**Respiration**	maximal respirational rate		−45.5	−35.9	−75.7	−49.8	−47.1	−44.4
	optimum temperature (°C)		33.0	34.1	35.4	33.8	35.4	34.2
	maximum temperature (°C)		45.4	47.1	45.4	44.6	49.1	45.5
	residual sum-square		1143	1062	3299	670	610	429
	temperature (°C) range for	optimal respiration (80% respirational rate)	25.3–39.1	26.1–40.4	28.8–40.4	26.9–39.1	27.0–42.1	26.9–39.8
		respiration (20% respirational rate)	12.5–44.3	12.7–45.9	14.6–43.4	14.6–43.6	132.−47.8	14.3–44.5

## Data Availability

Upon acceptance of the article, species sequences will be submitted to the National Centre for Biotechnology Information (NCBI) website.

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
