# Peer review of "Photosynthesis, Respiration, and Growth of Five Benthic Diatom Strains as a Function of Intermixing Processes of Coastal Peatlands with the Baltic Sea"

_microorganisms, 2022, doi:10.3390/microorganisms10040749_

Round 1

Reviewer 1 Report

REVIEW OF THE ARTICLE BY LARA REBECCA PRELLE AND ULF KARSTEN ENTITLED "PHOTOSYNTHESIS, RESPIRATION, AND GROWTH OF FIVE BENTHIC DIATOM STRAINS AS A FUNCTION OF INTERMIXING PROCESSES OF COASTAL PEATLANDS WITH THE BALTIC SEA" (microorganisms-1645895)

The Authors isolated and identified the strains of diatom microalgae from the Baltic Sea. They evaluated their growth preferences (depending on modification of the culture medium). Main subject of the work is the study of photosynthesis and respiration as a function of light photon flux density and temperature. Photosynthetic performance was measured as the rate of O2 emission. The work is in scope of the journal. The results are new and valuable. They were sufficiently discussed. Introduction is also good, in general. The work could be acceptable after a revision. My main criticisms are addressed mainly to the identification of algae and some methodological/terminological concerns (see below).

GENERAL COMMENTS

-Main concern of the work is the absence of the data on genetic idsentification of the strains. The Authors stated that it had been performed. These results should be added at least as supplementary materials. Obtained sequences should be submitted to the NCBI GenBank, GenBank accession numbers should be provided.

-Moreover the description of the identification is sufficiently described neither in [21] nor [29] cited by the Authors. Was the BLAST search performed. How was a phylogenetic tree reconstructed? It is impossible to identify the isolates without trees.

-Since morphology of the strains is also a matter of the studies, I suggest add representative photographs as supplementary materials.

-On the figures, the Authors used an approximation by a solid line. How did they get it? Please, explain. Assume also signifficance of these approximations, e.g. by r2 values.

-"as function of increasing photon fluence rates" - it is better to use the term "light curve".

SPECIFIC COMMENTS

l. 69-70. Biofilm is not a type of microorganisms. It is a spetial type of microbial communities. Please, rephrase.

l. 94. What is enhanced temperature? Higher temperature?

l. 104. What is "pore water"?

In Materials and Methods subsections have to be numbered.

Figure 1. Capture is incomplete.

l. 153-155. What do numbers in the brackets mean? GenBank IDs? Please, explain or remove.

For convenience, please indicate familu of Your algae in brackets at the first mention.

l. 166. Please, indicate the microscope (name, manufacturer).

l. 167. "in maximum of 0.05-1% of the diatom biomass" - i cannot understand it. One can estimate, for example, the percentage of cell number. DAPI staining gives fluorescence intensity and area but not mass.

l. 169. "with recent literature" - provide reference(s).

l. 185-187. This explanation is too confusing. It is better to say "219 µM NaNO3 (Carl Roth, Karlsruhe, Germany) as a source of inorganic nitrogen" instead of "nitrogen in a concentration of 219 µM (as NaNO3, Carl Roth, Karlsruhe, Germany)". Ther same is true about "phosphorus in a concentration of 27 µM (as NaH2PO4*H2O, Carl Roth, Karlsruhe, Germany)".

l. 187. Write ⋅ instead of *.

l. 190. "cultivated in always 15 ml" - what does it mean?

l. 192-197. This should be explained in more detail. Indicate spectral range of extitation and detection. Chlorophyll inrensity strongly depends on the physiological state of the cells, i.e. did you perform dark acclimation? Indicate, photon flux density of the excitating light. Indicate the type of an amplified photodiode (photomultipier?).

l. 192-197. Some phrases also have to be edited:

" Measurements proceeded every 24 h for 9 days byin a MFMS fluorimeter (Hansatech Instruments, King’s Lynn, United Kingdom). Using blue light LEDs emission (Nichia, Nürnberg, Germany) for the excitation of the chlorophyll a fluorescence, the resulting fluorescence was detected by an amplified photodiode and was separated from scattered excitation light through a long pass glass filter (RG 665; Schott, Mainz, Germany) and a bright-red gelatin filter (Lee, Brussels, Belgium)."

l. 201. N = N0 * e(µ * dt) - the expression should be edited. Do You actually mean N = N0eµdt?

It is better to write N(t) = N0eµt, where N(0)=N0, µ - specific growtrh rate.

l. 203. "N – fluorescence at the measuring day" should be "N – fluorescence intensity at the measuring day". Explain, also, what is No.

l. 204. "Growth rate" should be "specific growth rate".

l. 206. Acclimation to what?

l. 207. "Growth media composition for growth medium" - rephrase.

Table 1. In the caption, please, explain, what are 219 µM N and 27 µM P.

Table 1. Did You fix pH of each medium? pH Is a very important parameter, therefore it should be given.

l. 233-238. Presentation of equations is substandard. Please, see journal rules. What are "µg Chl a", E665, E750? What are their units? It should be explained. If "µg Chl a" is the chlorophyll a content in µg, the units in the left and right sides of the equation are not the same. please, explain untis. What do "83" means.

l. 261-261. Please, indicate manufacturers and countries of origin for all sowtware.

l. 273, 370. Avoid references in the results.

l. 271-277. This part must be completely reviesed. It is rather methods than results. Moreover, blast search is insufficient for species determination. Tree reconstruction is required.

l. 280-300, 431. days-1 are the units of the specific growth rate (not μ days-1).

l. 300. "in addition to N and P also Si" should be "in addition to NaNO3 and NaH2PO4 also Na2SiO3". The same is in l. 297.

Figure 2. Y axes should be named on each panel of the figure.

Figure 2, 3, 4, 5, 6. Explain, what is SD in the caption.

l. 310. The phrase sounds strange. There is no light-independent photosynthesis. What do you mean?

l. 310-325, 352-376,. Each pannel of each figure should be cited in the text separately. Otherwise, the panels should be condensed.

Table 2. In caption, please, explain, what do values and their errors mean.

l. 329. Remove -.

l. 352. "Effect of temperature on photosynthesis"?

Table 3, 4. What do ranges mean? Are there credibility intervals? Mean values?

l. 419-424. It is repetition of Introduction.

l. 467. double 'and' is confusing.

l. 467. I am not sure about existence of PO43- under environmental pH. It seems to be that HPO42- have to be more prevalent. Is it here actuay "inorganic phosphate species"?

l. 526. "... reverse reaction of diadinoxanthin de-epoxidation ... "?

l. 521-527. Although it is very important, other mechanisms, such as antioxydant production, ascorbate cycle and ROS-neutralizing enzymes cannot be excluded.

Author Response

Dear Reviewer

First of all, we like to thank you for the constructive and helpful comments as well as for the opportunity to carefully revise our manuscript.

We would first like to respond to your main concern with the species determination, which we are thankful you stated in depth!

We agree, that upon species determination, sequences should be deposited to NCBI GenBank. Therefore, the accession number is now included in the text (line 158) as requested. As we focused on morphological determination, which is a widely used and accepted approach on diatoms, we also agree with your suggestion to include LM and REM pictures of the respective species in the supplement (Hyalodiscus scoticus). REM pictures of the remaining four strains were already provided by Prelle et al. (2021).

To answer the questions of the BLAST: Yes, as described in the methods a BLAST search was performed in this publication for one isolate as well as in Prelle et al. (2021) who identified the remaining four isolates. 

For the species identification no phylogenetic trees were constructed, since the focus of our study was on ecophysiological traits. Nevertheless, we agree that phylogenetic trees are helpful with morphologically indistinguishable groups (e.g. green microalgae), but diatoms are known for many conspicuous morphological traits of their specific valves along with a comprehensive identification literature which makes species determination relatively easy. As mentioned in the acknowledgement, we discussed and matched species morphological determination with a  diatom taxonomist. However, to verify our morphological identification we additionally applied a genetical approach and matched the sequences using the NCBI BLAST function. With these commonly applied methods for diatom identification (which is now supported by SEM pictures in the supplement) and the match of sequences supporting our morphological determination we are are confident in our species determination.

Concerning the dotted lines within the figure 3, 4, 5 and 6, these were generated using the photosynthetic model of Walsby (1997) for the PI-curves using solver function by minimizing the sum of normalized squared deviations and the model of and of Yan and Hunt (1999) as referred to in the methods (lines 261-268). The respective confidence intervals were provided in the supplement material.

As suggested, we implemented the term “light curve” instead of “in response to increasing photon fluence rates”.

Specific comments

  • 69-70. We rephrased this sentence to “Both ecosystems inhabit many microbial communities including photosynthetic…” (in line 69)
  • 94. We rephrased this sentence to “…hence enhanced increasing water temperatures …” (in line 94)
  • 104. We added “…pore water within the sediment” to the sentence for better understanding (in line 104)
  • In Materials and Methods subsections have to be numbered. à we agree
  • Figure 1. We completed the figures caption
  • 153-155. We added “…with the respective GenBank IDs…” to the sentence (in line 152)
  • As requested we added the family of each species to the brackets (line 153-158)
  • 166. We indicate the microscope (name, manufacturer) (in line 168)
  • 167. We addressed this by changing “biomass” to “volume” (in line 170)
  • 169 We added the reference (in line 171)
  • 185-187. We agree and therefore included the reviewers suggestion (in line 187 - 189)
  • 187. We agree (in line 188)
  • 190. We deleted the “always” from the sentence as we think this might be responsible for the confusion (in line 192)
  • 192-197 We agree that in depth detail is relevant to keep track on the method, therefore we included a precise reference within the text to the originating study, explaining this method in depth. “…(see Karsten et al. [32] for further in-depth methodical details)” (in line 199-200)
  • 192-197. Thank you for this correction (in line 195-199)
  • 201, 203 and 204. We applied suggestions accordingly (in line 201-203)
  • 206. We added “acclimation to the experimental condition” (in line 206)
  • 207. We rephrased to “Media composition for experimental growth medium…“ (in line 208)
  • Table 1. We added “…and respective nutrient additions of f/2 + meatsilicate, inorganic nitrogen (N) as NaNO3 and inorganic phosphorous (P) as NaH2PO4⋅ H2O to supplement water basis” and the pH (in line 210)
  • 261-261. we added the respective manufactures and countries of origin for the software (in line 264-265)
  • 273, 370. We generally agree, however, in these cases the implemented references reinforce comprehension (in line 274)
  • 280-300. We applied your suggestion and defined “µ d-1” as specific growth rate (in line 201)
  • 300. in l. 297. We agree (in line 299, 304)
  • Figure 2. We added y axis label to all panels
  • Figures 2, 3, 4, 5, 6. We changed “SD” to “standard deviation” for better understanding (in line 308, 330, 342, 349, 385 and 393 )
  • 310. We agree and therefore changed this phrase to the widely used “PI-curve” for clear understanding (in line 313)
  • Table 2. We added “Data represent mean values ± standard deviation (n = 3, except Melosira nummuloides GM6 n = 4, Hyalodiscus scoticus GM6 n = 2)” to explain values and their errors (in line 329 - 330)
  • 310-325, 352-376 We addressed each panel separately
  • 329. Thank you, we removed the error.
  • 352. We agree, therefore we change the subheading to “Effect of temperature on photosynthesis and respiration” (in line 354)
  • Table 3, 4. We deleted “range of” to avoid confusion
  • 419-424 we added “As already mentioned in the introduction…” to indicate transition. (in line 421)
  • 467. We changed the word “and” to “to” (in line 466 and changed PO43- to “dissolved and hence bioavailable orthophosphate” (in line 466)
  • 526 We added “de-epoxidation” to the text (in line 530)
  • 521-527 We added “Along those other mechanisms such as the antioxidative potential (antioxidants and antioxidative enzymes) (Mittler 2002) have to be taken under consideration.” to this explanation (in line 531)

Kind regards

Reviewer 2 Report

This manuscript represents an interesting experimental approach to enhance our knowledge of the shallow coast ecocline, where and how is the terrestrial coastal zone influenced by marine processes, and more precisely biological processes. Microphytobenthos is an ecologically important surface sediment component. Its high primary production, role stabilizing sediment surfaces and food source for benthic animals are relevant components in shallow coastal waters. Experimental approaches such as the one presented in this work can help to gain knowledge about the future of these systems under the projected sea level change scenarios.

General concept comments

The manuscript is clear, relevant for the field and presented in a well-structured manner. The abstract summarizes perfectly the work, including the question investigated, the methods used, the principal results and conclusions. The introduction provides sufficient context and background for the reader to understand and evaluate the research and includes a short history or relevant background that leads to a statement of the problem that is being addressed.
Discussion and conclusions are well related to the goals of the study, as stated in the introduction, and properly analyzed to the available knowledge. Conclusions are well-supported by the findings being neither too narrow nor too broad.
In any case, there are some aspects of the work that I would like to consult with the authors. It is not at all obvious why only the results obtained on GM6 and GM9 media are compared. This should be clarified, preferably in the results section. Likewise, the different levels of significance obtained in the tests are labeled with lowercase and/or uppercase letters representing significant levels among all means as calculated by a one-way ANOVA. But it is not self-evident and a clarification would be helpful.
It would also be interesting to know the typical composition of the diatom communities in both environments and the relevance of the selected strains.

Specific comments

Line 36: I do not have data for the Baltic Sea, but in other seas the rise in sea level due to thermal expansion is a factor to take into account
Line 86: I don't think it's the right way to express it. The ability to move in benthic diatoms is not a 'response' to prevailing conditions. The environment selects.
Line 127: Any specific corer?
Line 185: What is the justification for choosing this nutrient ratio?
Line 214 and 246: Why GM6 and GM9? Explain.
Line 473: Could you give a value for SiO2?

Author Response

Dear Reviewer

First of all, we like to thank you for the constructive and helpful comments as well as for the opportunity to carefully revise our manuscript.

We agree that it would be interesting to know the typical species composition. Unfortunately, so far no comprehensive studies addressing this important aspect were done at the study sites.

We further clarified the meaning of the lowercase and capital letters to the statistical analysis. “…post-hoc Tukey’s significant differences test (critical p-value < 0.05). Significant differences were indicated by lowercase and capital letters.“ (line 268-269)

Specific comments in the manuscript

  • Line 36: Thank you for this hint. We found data on thermal expansion and added these to the text “As a consequence of glacial melting and thermal expansion [3], the sea level of the Baltic Sea is continuously increasing by 2 mm yr-1 in the southern Baltic Sea [4] which…” (in line 36-37)
  • Line 86: This statement is based on the citation of Cohn et al. 2021 using the same vocabulary. As we did not investigate motility we are unable to contradict or to agree with the word “response”. Hence we would prefer to keep this statement.
  • Line 127: As the cores were taken by hand, no corer was mentioned.
  • Line 185: This was stated in the following part of the sentence “…to mimic the highest nutrient values found in the sampling site of “Heiligensee und Hütelmoor” (line 191)
  • Line 214 & 246: We added an explanation. “These two media were chosen to investigate photosynthesis in likely occurring media changes after a storm surge events. The medium of GM2 was excluded due to partially no growth response in the growth experiment.” (in line 218-219)
  • Line 473: We added the concentrations to the text “…(Baltic Sea <1.0 SiO2 mg l–1, peatland >6.0 SiO2 mg l–1; using the…”  (in line 477)

 Kind regards

Reviewer 3 Report

I am very confused after reading the manuscript. Authors proposed very interesting topic with important analysis of diatom growth studies but same the basics of studies have fundamental deficiencies.

Authors works with cultures of diatom and put forward far-reaching conclusions based on practically unidentified species ! Even after finishing diatom cultures Authors have no identified strains to species level. For  any consideration about ecology and environmental requirements, the proper species identification in one of the most important and fundamental aspects of studies.

Authors should identify the structure of diatom assemblages in both environments, should perform water analysis and select proper species (base on domination structure) for culture. If these steps are not  carried out properly the rest of studies not make sense !

In my opinion without this data paper should be rejected, but if Authors will be able to identify studied species (and will prove it with photos) the paper can be publish after review.

Same small additional comments in the ms file.

Best regards

Author Response

Dear Reviewer

First of all, we like to thank you for the constructive and helpful comments as well as for the opportunity to carefully revise our manuscript.

We would first like to respond to your main concern with the species determination and the structure of diatom assemblages.

We fully agree that the species determination is an important aspect in our study. Nevertheless, we disagree, that our study is based on “practically unidentified species”. Diatom species used in this study were morphologically identified and backed up with molecular analysis which is a commonly applied procedure for diatoms identification (see our response to reviewer 1). Four of the applied diatoms were already identified by Prelle et al. (2021) providing REM images as well as molecular data, of which accession numbers were deposited in NCBI GenBank. An additional strain was added (Hyalodiscus scoticus) for this study. This species was identified morphologically and also supported with molecular data. As the reviewer kindly pointed out microscopic pictures of this species were missing. To make this more coherent, we added REM and light microscopic images of the latter species to the supplement and also deposited molecular sequences to NCBI GenBank, which we referenced using the accession number (in line 158). We also fully agree that knowledge on the structure of diatoms assemblages in the studied environments is very important. However up to now such data are missing  for the investigated habitats.

Specific comments in the manuscript

  • Line 153: The cultures were chosen as they already provided data on ecophysiological response patterns and are therefore valuable for in depth investigation for the effect to changing/mixing habitat water bodies.
  • Line 186: Yes, nitrogen and phosphorous concentrations were based on field measurements as reference in line 191-192.
  • Line 271: This is a great point which we would have gladly liked to address as well. However, no data on the abundance or general species composition of diatoms have been done in this nor any close areas. Nevertheless, all isolates were abundant in the samples upon isolation.
  • Line 272: Yes, the authors and a diatom taxonomist that was approached were not able to identify two isolates to a species level, despite the use of SEM pictures (these are included in the referenced publication Prelle et al. 2021, SEM images of Hyalodiscus scoticus are now included in the supplement). For sure the genus Planothidium needs taxonomic revision in the future.
  • Molecular data are available for all five strains of which four were previously identified. All sequences were deposited in NCBI GenBank with the corresponding accession numbers in line 154-158.
  • Line 472: The investigated Baltic Sea next to the study sites consists of 1/3 sea water and 2/3 freshwater and therefore does not reflect a fully marine habitat, but rather a brackish system. Nevertheless, we added our measured SiO2 concentrations to support our argument. (line 477)
  • Line 554: we rephrased this sentence to “More importantly, our data clearly show that peatland water enhances growth with the indication of heterotrophic growth”, to underline the respective results (in line 559).

Kind regards

Round 2

Reviewer 1 Report

I agree with the responses to most of my comments and most of changes made in the manuscript. In general it has been improved, however there are still some important issues, which have to be removed before final decision on the manuscript.

1) You must add references to each supplementary file in the corresponding places of the text. In the end of the manuscript supplementary materials have to bi listed with their short description: Supplementary file S1. ..., Supplementary file S2. ...

2) l. 168. Indicate also, that Your "algae were studied by scanning electron miroscopy (SEM) on the microscope XXX. Sampals for SEM were prepared as preciously described [Add the reference for sample preparation]".

3) l. 245. Temperature depenDENCE of photosynthesis and respiration.

4) l. 201 nd further through the text. On my original comment on growth rate (r): the growth rate is defined as changing of culture density (N) in time (t), i.e.

r=dN(t)/dt,

and has units of density units/days; density units are cell number, dry mass, etc.; in Your case it is fluorescence intensity units, whereas µ in Your expression is the specific growth rate defined as

µ=1/N·dN(t)/dt=dln(N)/dt,

its units 1/days. Please, feel the difference between growth rate and specific growth rate and modulate it through the text.

5) My original comment: "Figure 2. Y axes should be named on each panel of the figure" - it has not been done in the text.

6) My original comment: "l. 310-325, 352-376,. Each pannel of each figure should be cited in the text separately. Otherwise, the panels should be condensed." It has been done only for Figure 2. But the same is true for other figures with pannels (3-8). You can cite each panel easely near each strain in the text.

7) "blue light LEDs emission" - the phrase has not been edited according to suggestions. Please, understand, that LEDs are light emitting diodes, therefore in ful you wil obtain "blue light light emitting diodes emission", which sounds strange, therefore the correct form is "blue LEDs (Nichia, Nürnberg, Germany)".  

8) Considering my original comment on cureves fitting on figs 3-6: I understand your response, but it should be clearly stated in the text, i.e. in the methods section. Moreover, you stated, that  "the respective confidence intervals were provided in the supplement material". I did not find it in supplementary materials, microscopic photographs only.

9) One more issue about the statement about the carotenoid de-epoxidation: "Further, an accumulation of the photoprotective xanthophyll pigment diatoxanthin via the reverse reaction of diadinoxanthin (de-epoxidation) weakens the oxidative stress under high light conditions" (l. 529-531). The statement still requires correction. Why "de-epoxidation" is in brackets?  In addition, de-epoxidation is not a reversible reaction: de-epoxidation requires Monogalactosyldiacylglycerol and ascorbate, whereas epoxidation requires O2. Thus epoxidation and de-epoxidation are two different reactions. The correct form could be "Further, an accumulation of the photoprotective xanthophyll pigment diatoxanthin via the reaction of diadinoxanthin de-epoxidation weakens the oxidative stress under high light conditions".

Author Response

Dear Reviewer

Thanks for the thorough checking of our manuscript.

1) We applied this according to your suggestion.

              “In the supplement table confidence intervals of the modeled data are shown,…” Line 269

              “SEM (scanning electron microscopy) and light microscopy … are deposited in the supplement figures.” Line 169

2) We added the references on preparation and methods “SEM (scanning light microscopy) and light microscopy images were prepared following the same methodological approach as Prelle et al. (2021) and are deposited in the supplement figures.” Line 169-171

3) we changed this according to your suggestion. Line 245

4) we are fully aware on the difference between “growth rate” (biomass per time) and “specific growth rate” (biomass per time, BUT initial biomass is considered as well). Only the latter allows direct comparison between strains. We clarified the whole text by always writing “specific growth rate” to avoid any misunderstanding.

5) Thank you for this hint!

6) We followed your suggestion, as far as it is applicable and without overcrowding the text.

7) We changed this according to your suggestion. Line 202

8) We think that the methods of the fitting are clearly stated in the methods (see section and 2.4 light response curves (line 240), 2.5 temperature dependence of photosynthesis and respiration (line 257) and 2.6 statistical analysis (line 260), as well as in the respective figures 3, 4, 5 and 6. Nevertheless, we can not understand, why you were not able to receive the supplement material with the confidence intervals as those were supplied during the first submission. Therefore, we will attach it to the specific response again.

9) We agree and therefore we adjusted the text to you suggestion. Line 532

Kind regards

Reviewer 3 Report

Dear Authors, Dear Editorial Office

thank you for taking into account my comments. In my opinion Authors should resign from 3 plates of H. scoticus (one plate will be enough), and instead of include all studied species. I know that they use the same taxa like in Prelle et al 2021, but even in cited paper they did not present high quality images, so they should do this here.

After inluding pictures of all studies species, paper will be suitable to publish. 

Best regrds

Author Response

Dear Reviewer

Thank you for your comment.

We would like to keep the 3 plates of H. scoticus, nevertheless we added a plate with SEM images for the remaining isolates from Prelle et al. 2021.

Kind regards